# From Motion to Causation: The Diachrony of the Spanish Causative Constructions with *traer* ('Bring') and *llevar* ('Take')

**Julio Torres Soler** [1,*] and **Renata Enghels** [2]

1   Department of Spanish Studies, General Linguistics & Literature Theory, University of Alicante, 03690 Alicante, Spain
2   Department of Linguistics, Ghent University, 9000 Ghent, Belgium
*   Correspondence: julio.torres@ua.es

**Abstract:** This paper describes the historical evolution of the Spanish causative micro-constructions with the motion verbs *llevar* ('take') and *traer* ('bring') (e.g., *el miedo llevó al ladrón a cometer un error*, 'the fear caused the thief to make a mistake'). In order to reconstruct the historical development of these micro-constructions between the 13th and 20th centuries, all causative uses of *llevar* and *traer* were extracted from the *Corpus del Diccionario Histórico*. This corpus was annotated for a series of formal and semantic parameters that count as indexes of grammaticalization, and was submitted to a quantitative productivity analysis. The results point to the existence of a subschema formed of verbs of caused accompanied motion, which has semantically specialized in the expression of indirect causation. From a formal point of view, this subschema is characterized by a low level of syntactic incorporation of the causative verb and the infinitive. In addition, it is shown that the productivity of the causative micro-constructions under study is determined by semantic changes experienced by *llevar* and *traer* as full lexical verbs during the history of Spanish. The late development of the micro-construction with *llevar* is explained by the initial tendency of this verb to express motion events not bounded by an endpoint. From the 16th century onwards, the decline in the micro-construction with *traer* and the rise in the micro-construction with *llevar* results from the consolidation of the deictic meaning of the verb pair.

**Keywords:** grammaticalization; diachronic construction grammar; causation; causative construction; *llevar*; *traer*





## 1. Introduction

From a typological point of view, causation can be expressed by different linguistic procedures, including lexical, morphological and syntactic mechanisms (Shibatani and Pardeshi 2002). One way of expressing causation in Spanish is through the well-known causative construction (e.g., *El maestro hizo disculparse al alumno* 'the teacher made the student apologize'). From a Construction Grammar (henceforth CxG) perspective, the causative construction can be understood as an abstract schema that links a set of formal and functional features. From a semantic perspective, the first participant, i.e., the causer, acts upon a second participant, i.e., the causee, who is involved in a concatenated event, referred to as the effect or caused event. It is, therefore, a construction composed of two linked events: a causing event and a caused one. From a formal viewpoint, the causative construction is made up of the following elements: [nominal phrase 1 + causative verb + nominal phrase 2 (+ preposition) + infinitive].[1]

This overarching causative schema is realized through several micro-constructions which share inherited features, but also present formal and semantic differences. The micro-construction that has received most attention is the one with the causative verb *hacer* ('make') (among others, Cano-Aguilar 1977; Davies 1995; Roegiest and Enghels 2008; Vivanco 2019), defined as the prototypical causative micro-construction due to its

high frequency and semantic versatility (Enghels and Comer 2020). More recently, some attention has been paid to the causative micro-constructions with *dejar* ('let') (Enghels and Roegiest 2014; Maldonado 2007) and with *poner* and *meter* ('put') (Comer 2020; Enghels and Comer 2020; Vivanco 2020). Other Spanish causative verbs, such as *obligar* ('force') or *inducir* ('induce'), with some exceptions (Alfonso Vega 1997), have received little attention in the literature.

The syntactic relationship between the main causative verb and the subordinate infinitive in the causative construction has been debated in the literature. A good number of researchers consider it to be a biclausal structure, assuming that the infinitive and its complements form a phrase subordinate to the causative verb (Alfonso Vega 1997; RAE 2009; Treviño 1994; Vivanco 2019), while others opt for a monoclausal or periphrastic analysis, in which the causative verb functions as an auxiliary[2] (Comer 2020; Enghels and Comer 2020; Hamplová 1970; Kemmer and Verhagen 1994). In fact, the degree of syntactic incorporation between the causative verb and the infinitive varies between Romance languages (Da Silva 2012) and also within the same language, depending on its evolutionary stage (Davies 1995) or the main causative verb (Comer 2020; Enghels and Comer 2020). For example, Enghels and Comer (2020), in their contrastive study of causative micro-constructions with *hacer* (e.g., *El maestro hizo disculparse al alumno* 'the teacher made the student apologize'), *meter* (e.g., *Metió a su hijo a trabajar* '(s)he put his child to work') and *poner* (e.g., *El presidente puso a debatir a los ministros* 'the president set the minsters to debate'), observe that the variant with *hacer* presents the highest degree of syntactic incorporation of the three.

In the same study, the authors observe that the causative micro-constructions with *meter* and *poner* present a more restricted semantic profile than the one with *hacer*: they are preferably used to designate events of direct causation, express an inchoative aspectual value and are frequently associated with infinitives belonging to the culinary domain (Enghels and Comer 2020, p. 190). As a consequence, the micro-constructions with locative verbs are defined as a particular subschema of the causative construction, in which particular syntactic and semantic features from their locative origin persist. Finally, specific variation among the causative micro-constructions has been related to their stage of grammaticalization. Based on parameters such as the syntactic incorporation between the causative verb and the infinitive and the diversity of semantic contexts they admit, Enghels and Comer (2020) conclude that the micro-construction with *hacer* is more grammaticalized than the micro-constructions with *poner* and with *meter*. Moreover, from a diachronic perspective, the authors observe that the micro-construction with *poner* gradually undergoes an advance in its grammaticalization, unlike the micro-construction with *meter*, whose evolution shows a certain degrammaticalization and a semantic specialization towards increasingly restricted contexts of direct causation and material caused events. Against this background, the present paper provides a diachronic analysis of the understudied causative micro-constructions with the verbs *llevar* 'take' and *traer* 'bring'. From a semantic viewpoint, the micro-construction with *llevar* has been classified within the scope of indirect causation (Da Silva 2004, 2012), which implies that the causer has little control over the caused event, performed by the causee autonomously. Formally, the causative micro-constructions with the verbs *llevar* and *traer* are identified by the preposition *a* before the infinitive, which distinguishes them from other causative micro-constructions that use the verb *hacer*. Thus, the formal structure of these causative micro-constructions is the following: [NP1 + *llevar*/*traer* + NP2 + *a* + infinitive], as shown in the examples in (1), taken from the *Corpus del Diccionario Histórico* (CDH).

(1) a. El espíritu de intenso lucro que animaba a la economía liberal ***llevaba*** *al patrono a imponer* jornadas agotadoras. (Carlos García Oviedo, *Tratado Elemental de Derecho Social*, 1946).

> 'The spirit of intense profit-making that animated the liberal economy **led** the employer to impose strenuous working days'.

b. El descubrimiento de esta lápida *nos* ***trae*** *por la mano á tratar* de los tesoros depositados en Guarrazár. (Vicente de la Fuente, *Historia eclesiástica de España, II*, 1855–1875).

'The discovery of this tombstone **leads** us by the hand to deal with the treasures deposited in Guarrazár'.

At first glance, the two micro-constructions appear very similar. Still, in order to better understand the development of the causative construction, this study attempts to describe the formal and functional characteristics of the micro-constructions with *llevar* and *traer*, and to compare their degree of grammaticalization in different stages of Spanish. More specifically, we will try to find out if they experience an advance in their grammaticalization over time, as was the case of the micro-construction with *poner*, or if there is a semantic specialization in increasingly restricted contexts, such as *meter*. On the other hand, as full verbs, *llevar* and *traer* express events in which an agent moves a figure in a controlled way by the translocation of the agent's body (Talmy 2000, p. 51). Given the semantic relatedness of *llevar* and *traer* as motion verbs, the question arises whether the causative micro-constructions with *llevar* and *traer* are part of a particular subschema, different from that of other causative micro-constructions previously mentioned.

Moreover, we examine the relationship between the historical development of the semantics of *llevar* and *traer* as full verbs and their use in the causative construction. This is in line with the outcomes of previous studies suggesting that the diachrony of grammaticalized verbs can be influenced by changes at the lexical-semantic level, and that both grammaticalized and full lexical meanings of verbs are strongly interrelated in polysemous semantic networks. Garachana Camarero and Hernández Díaz (2020), for instance, show that the emergence of the verbal periphrases <*tener de/que* + infinitive> in Old Spanish occurs by analogy with <*haber de/que* + infinitive> as soon as *tener* had developed its possessive meaning as a full verb, becoming a near-synonym of *haber*. Moreover, the authors show that the periphrases <*haber de/que* + infinitive> have semantically narrowed in the 16th century, as a consequence of the loss of the possessive meaning of *haber* in favor of its existential meaning.

The remainder of the paper is organized as follows. Section 2 focuses on the origin of the causative micro-constructions with *llevar* and *traer*, which leads us to explore the meaning of *llevar* and *traer* as motion verbs and in related metaphorical uses. Section 3 provides more details on the diachronic corpus that serves as an input for the analysis of the causative micro-constructions with *llevar* and *traer*. Section 4 presents the results and describes the outcomes of a productivity analysis of the objects of study, as well as their formal and semantic characteristics. Section 5 further discusses the results and presents the conclusions of this work.

## 2. From Directed Motion to Causation

In their prototypical meaning, *llevar* and *traer* can be classified as Caused Accompanied Motion verbs (CAM verbs).[3] CAM verbs express events in which an entity causes the change in location of a second entity in a particular way: (a) the event causing the movement has a prolonged duration and is simultaneous to the movement itself, and (b) the agent accompanies the displaced entity throughout the motion event, changing his location along with it (Margetts et al. 2022, p. 936). What differentiates *llevar* 'take' and *traer* 'bring' in present-day Spanish is their component of spatial deixis: *traer* is a coming verb, meaning that it expresses motion towards the deictic center, while *llevar* is a going verb that expresses motion towards places other than the deictic center. However, importantly, this semantic opposition is not constant throughout the diachrony of Spanish. It has indeed been shown that, in Old Spanish, the use of *llevar* and *traer* as motion verbs is not always defined by deictic factors, but that lexical aspect or *Aktionsart* can also play a role (Torres Soler 2023).[4]

In all diachronic stages, the verb *llevar* is used to express motion towards the place occupied by the receiver of the communicative situation (2a) or to a place different from that occupied by both the interlocutors (2b).

(2) a. Otrosi del libro que me enuiaste a mandar que uos **leuasse** sabet sennor que non lo tengo aqui mas yo he enbiado por el e **leuar** uos lo he quando me fuere para uos. (*Colección diplomática de D. Juan Manuel*, 1308).

'Moreover, about the book that you ordered me to **bring** you, you must know, lord, that I do not have it here, but I have sent someone for it and I will bring it to you when I will come to you'.

b. Se ha fecho y fase en la dicha çibdad de Xeres vna feria, a la qual se **lieua** e descargan muy muchas mercadurías, las quales se **lieuan** de allí a otras partes. (*Tumbo de los Reyes Católicos del concejo de Sevilla*, 1479).

'It has been done and it is done in the aforesaid town of Jerez a fair to which many goods are **taken** and unloaded, which are **taken** from there to other places'.

In Late Latin, *levare* (the etymon of *llevar*) functioned in one of its meanings as a verb of removal (Cifuentes Honrubia 2008) that expressed movements away from the initial location, without a specific Goal bounding the event at its endpoint. These kinds of movements are referred to as *ablative* or Source-oriented motion events, as opposed to *allative* or Goal-oriented motion events, which consist in moving towards a final location (Kopecka and Vuillermet 2021; Stefanowitsch and Rohde 2004). In Old Spanish (12th–15th centuries), although *llevar* also accepted delimited Goals, it still showed a tendency to express Source-oriented motion events. For instance, in (3), *llevar* is used to refer to displacements bounded by their Source, which is a specific place (*el prado* 'the meadow'), but not by their endpoint, which it is not specified, because it is not relevant to know where the flowers ended up.

(3) Los omnes e las aves, quantos acaecién, **levavan** de las flores quantas **levar** querién, mas mengua en el prado niguna non facién, por una qe **levavan** tres e quatro nación (Gonzalo de Berceo, *Los Milagros de Nuestra Señora*, 1246–1252).

'Men and birds, as many as came along, **took** as many flowers as they wanted to **take**, but they did not cause any damage to the meadow; for each one they **took**, three or four were born'.

The verb *traer*, for its part, was used in Old Spanish (12th–15th centuries) to express movements with any deictic orientation, meaning that it could be used to express motion events independently of the place occupied by the interlocutors. However, in the 16th century, the deictic component of its meaning became established, and therefore, in Classical Spanish (16th–17th centuries), it could only be used for motion events towards the area of the speaker or the receiver. In Modern and Contemporary Spanish (18th century-present), the meaning of *traer* further specialized in the expression of motion towards the place of the speaker. The following are examples in which *traer* expresses motion towards the speaker (4a), the receiver (4b) and towards a place different from that occupied by both interlocutors (4c) in Old Spanish.

(4) a. Ydvos para los joyeros e **trahedme** composturas. (*La historia de la donzella Teodor*, 1250).

'Go to the beauticians and **bring** me cosmetics'.

b. Vi vuestra carta en que me enviastes dezir que, por vos fazer yo bien e merçed, tove por bien e mande que ningun omne non **troxiesse** a vuestra villa por dos annos vino de fuera parte. (*Documentos de Alfonso X dirigidos al Reino de León*, 1271).

'I saw your letter in which you told me that, in order to favour you, I decided and ordered that nobody would **bring** wine from outside to your village for two years'.

c. Vuestros subditos, [ . . . ], los quales deziys tienen licencia de nuestro Sancto Pedre de negociar e **traher** madera, pez e otras mercadurias a las tierras de los turcos e de los egipcianos. ("Fernando al cardenal maestre de Rodas [ . . . ]", *Documentos sobre relaciones internacionales de los Reyes Católicos, II*, 1496).

'Your subjects, [ . . . ], who you say that they have licence from our Holy Father to negotiate and **take** wood, fish and other goods to the lands of the Turks and the Egyptians'.

As regards its Aktionsart, *traer* in Old Spanish functioned as an allative or Goal-oriented motion verb that specialized in the expression of motion events with a specific Goal bounding the event at its endpoint. For instance, in (5), *traer* expresses a motion event that is bounded by the Goal *al dicho monasterio* ('to the said monastery').

(5) Ellos siendo llamados algunas veses por los dichos abad e convento sus señores, o por su mandado, que no quisieron **traer** el pan e la sal con sus bestias <u>al dicho monesterio</u> (*Colección documental de Alfonso XI*, 1347)

> 'They, being sometimes called upon by the said lords abbot and members of the monastery, or by their command, did not want to bring the bread and salt with their beasts <u>to the said monastery</u>'.

Therefore, in Old Spanish, *llevar* and *traer* competed in the expression of motion towards places not occupied by either of the interlocutors, and the choice of one or the other was influenced by their lexical aspect. In these contexts, Source-oriented movements without a concrete Goal are mostly expressed by *llevar*, while *traer* is used preferably in contexts where a specific Goal bounds the motion event. In example (6), *llevar* and *traer* express motion to places not occupied by the interlocutors. *Traer* is employed when there is specific Goal (*a la dicha feria* 'to the said fair') bounding the event at its endpoint, while *llevar* is chosen when the Source is a specified place, namely the same fair, and the Goal is not concretized.

(6) E a todo lo que consigo **truxieren** <u>a la dicha feria</u> e toiueren en ella e **lleuaren** <u>della</u> (*Tumbo de los Reyes Católicos del concejo de Sevilla*, 1479).

> 'And everything that they **carry** with them <u>to the said fair</u> and that they have in there and **take** away <u>from there</u>'.

The aspectual differences of *llevar* and *traer* in Old Spanish were progressively weakened throughout the Middle Ages in favour of their incipient deictic value, until *llevar* became the only verb capable of expressing motion to a place different from the deictic center in the 16th century. In short, three stages can be distinguished in the history of *llevar* and *traer* as motion verbs: (a) in Old Spanish (12th to 15th centuries), their deictic opposition was not yet fully consolidated, so their distribution depended on both incipient deictic factors and Aktionsart differences, (b) in Classical Spanish (16th to 17th centuries), their opposition was already fully deictic and the deictic range of *traer* comprised both the 1st and 2nd person, and (c) in Modern Spanish (18th to 20th centuries), their opposition was still fully deictic and the deictic range of *traer* was restricted to the first person. Parallel to their meaning as motion verbs, *llevar* and *traer* developed a series of metaphorical extensions. In concrete, we hypothesize that the causative micro-constructions under study derive from metaphorical expressions that indicate the causation of a process as a result of which the direct object gets involved in a new event. This event, which can be either static or dynamic, is mentioned by means of a nominal expression preceded by a preposition (usually *a*). It is metaphorically conceptualized as an abstract place reached by the direct object, thus as the Goal of a metaphorical motion event. The underlying conceptual metaphor, as formulated by Lakoff and Johnson (1999), is CAUSATION IS FORCED MOVEMENT. On the one hand, in the source domain of caused motion, we typically find an agent (NP1) that exerts a physical force to displace a figure (NP2) that receives the driving force and consequently moves to a new location (<*a* + NP3>). An example is *María llevó el jarrón al salón* ('María took the vase to the hall'). On the other hand, in the target domain of causation, we find a causer (NP1), animate or not, that causes a causee (NP2) to be involved in an event (<*a* + NP3>) (7).

(7) a. El uno de tus amigos es aquel que te tuelle del mal y te **lieva** <u>al bien</u> (*Libro de los buenos proverbios que dijeron los filósofos y sabios antiguos*, 1250)

> 'Your friend is the one who takes you out of evil and **leads** you <u>to good</u>'.

b. Qvando la sacra excelencia nuestra yglesia começo vnos **truxo** con clemencia y otros <u>a la penitencia</u> (Pedro de Salinas, *Poemas*, 1533).

'When the sacred excellence of our Church began, it **led** some with mercy and others to penance'.

In examples (7a–b), the nominal expressions preceded by *a* no longer represent the Goal of a displacement in space, but an event in which the direct object gets involved. In example (7a), *al bien* ('to good') is al state, whereas in (7b), *a la penitencia* ('to penance') is a dynamic event.

The causative micro-constructions with *llevar* and *traer* originate from the aforementioned metaphorical expressions, when the event in which the direct object gets involved is no longer expressed by a nominal expression, but through an infinitive. The close conceptual relationship between these metaphorical expressions and the infinitive construction is corroborated by examples such as (8), which is the oldest documented example of the causative micro-construction with *llevar*.

(8) Sean vuestras cobdicias **levadas** a aver buena fama; non las **levedes** a las malicias nin a las feas cosas. (*Bocados de oro*, 1250).[5]

'Let your ambitions be **led** to have a good reputation; do not **lead** them to evil and ugly things'.

Example (8) illustrates a caused event expressed by the infinitive and its complement *aver buena fama* ('to have a good reputation'), which is opposed in the following sentence to the abstract locative complement *a las malicias nin a las feas cosas* 'to evil and ugly things'. Both sentences imply that the direct object is caused to get involved in an event, but only in the second this event is transparently conceptualized as an abstract Goal. In short, the origin of the causative micro-constructions with *llevar* and *traer* can be summarized as follows: (a) 'to lead (*llevar/traer*) to a concrete place' >, (b) 'to lead (*llevar/traer*) to an abstract place' >, (c) 'to lead (*llevar/traer*) to an event'.

In light of the semantic changes experienced by *llevar* and *traer* as full lexical verbs throughout the history of Spanish, the following sections explore whether they had an impact on the use of *llevar* and *traer* in the causative construction. More specifically, it is examined whether the evolving semantic differences between the verbs had an impact on the overall evolving productivity and varying degrees of grammaticalization of the causative micro-constructions under study.

## 3. Methodology

### 3.1. Data Collection

For the analysis of the causative micro-constructions, relevant data were extracted from the *Corpus del Diccionario histórico de la lengua española* (CDH), an extensive diachronic corpus of Spanish elaborated by the Real Academia Española.[6] More specifically, we used two of the three query layers of the CDH, namely the *nuclear CDH* layer, consisting of texts from all periods up to the year 2000, characterized by their strict philological rigor, and the *S. XII-1975* layer, which allowed us to reach a broader view of the construction, thanks to the large number of historical documents included. However, the *1975–2000* layer was not used because the number of texts that it contains from a 25-year period is too large for the macro-diachronic interests of this paper. Since the focus is not on diatopic variation, only data from Spain were considered.

We downloaded all occurrences of the lemmas *llevar* and *traer* (in any of their forms) followed on the right by the preposition *a* and an infinitive. We included cases in which the causative verb and the <*a* + INF> group are contiguous as well as cases in which they are separated by up to 3 intercalated linguistic elements.[7] In a next phase, the data were manually checked in order to exclude all non-relevant cases from the dataset. As a result, we obtained a corpus of 1188 tokens of the causative micro-construction with *llevar* and 218 tokens with *traer*.

*3.2. Parameters of Analysis*

To begin with, the internal dynamics of the causative construction is explored through a productivity analysis (Section 4.1). The productivity analysis centers around token frequency, which measures the general frequency of the micro-constructions, and type frequency, which accounts for their semantic spread and density (Barðdal 2008). It is particularly interesting to investigate (a) to what extent both micro-constructions with *llevar* and *traer* have competed in the different stages of Spanish, and (b) whether their degree of productivity was affected by the semantic changes experienced by *llevar* and *traer* as full lexical verbs.

Second, in order to account for the degree of grammaticalization of the causative micro-constructions with *llevar* and *traer* in the different stages of Spanish, the dataset has been annotated according to a series of well-defined formal and semantic parameters. In concrete, four formal parameters are analyzed with a focus on the fixation of the construction and its degree of unithood. First, in order to study the degree of formal fixation, the type of preposition introducing the infinitive complement will be examined. In fact, in addition to the preposition *a* ('to'), structural variants with the preposition *en* ('in') and without prepositions have been attested in the corpus. This observation raises the question whether a process of morphosyntactic specialization (in terms of Hopper 1991) or obligatoriness (in terms of Lehmann 2015) has occurred, which would indicate further progress in the grammaticalization process (Section 4.2.1).

Next, the notion of unithood refers to the conceptualization of the causative situation as a single action, rather than two autonomous events. A high degree of unithood relates to a higher degree of grammaticalization of the construction (Enghels and Comer 2018). The degree of unithood can be observed through the degree of incorporation or syntagmatic linkage between the causative verb and the infinitive. This criterion is operationalized by looking into the presence or absence of intercalated linguistic elements between the causative verb and the infinitive, in concrete the causee (Section 4.2.2) and other adjuncts (Section 4.2.3). A fourth criterion that provides further insight into the degree of unithood of the construction is the degree of association between the causative verb and the infinitive (Comer 2020; Torres Cacoullos 2000). While the combination of the causative verb with a single infinitive indicates a high degree of association between the two, the presence of multiple infinitives indicates a lower degree of association, and consequently, a lower degree of grammaticalization (Section 4.2.4).

Finally, in order to comprehend the functional development of the causative construction, two semantic parameters are examined. First, in grammaticalization processes, the elements of the construction undergo a loss of semantic integrity (Lehmann 2015, pp. 126–28). In the case of verbal grammaticalization, this process takes the form of a relaxation of the semantic constraints operating in the selection of the verbal complements (Lamiroy 1994, p. 35). Moreover, from a CxG perspective, it has been observed that, as a constructional schema develops and becomes more abstract, the construction is expected to expand semantically to new domains, thus increasing the repertoire of linguistic units with which it can be combined (Barðdal 2008; Enghels and Comer 2018). In order to investigate if the causative micro-constructions with *llevar* and *traer* enhanced their semantic scope over time, the animacy of both the causer and the causee (Section 4.3.1) and the semantic types of the infinitives (Section 4.3.2) are examined.

## 4. Results

*4.1. General Productivity Measures*

Linguistic units involved in a grammaticalization process tend to experience an increase in their token frequency (Traugott and Heine 1991, p. 9). Similarly, diachronic CxG studies point out that constructionalization and constructional changes coincide with a rise in token frequency (Traugott and Trousdale 2013). Consequently, an increasing token frequency of the micro-constructions with *llevar* and *traer* might be an indicator of their grammaticalization and a sign of the further development of the causative construction

as a schema. While absolute token frequency counts the total number of occurrences of the micro-constructions in the corpus, the measure of relative token frequency shows their standardized frequency (e.g., per million words).

Next, type frequency accounts for the number of different lexical items that can fill an open slot in the construction (Barðdal 2008), in our case, the infinitive slot. When a construction is involved in a grammaticalization process, it is expected to expand to new contexts, to increase its productivity and thus to enlarge its collocational range (Traugott and Trousdale 2013). Consequently, an increase in the type frequency of the causative micro-constructions with *llevar* and *traer* should be interpreted as a sign of the expansion of the construction to new semantic domains. Again, it is informative to take into account both the absolute type frequency, which indicates the total number of different infinitives, and the relative type frequency, which measures the number of infinitives documented per million words. Table 1 shows the token frequency and the type frequency of the causative micro-constructions with *llevar* and *traer*, both in absolute and relative numbers, in the period between the 13th and 20th century.

**Table 1.** Token and type frequencies of the causative micro-constructions with *llevar* and *traer*.

|  | 13 | 14 | 15 | 16 | 17 | 18 | 19 | 20 |
|---|---|---|---|---|---|---|---|---|
| | | | | Absolute token frequency | | | | |
| *Llevar* | 1 | 1 | 9 | 48 | 49 | 19 | 254 | 807 |
| *Traer* | 6 | 13 | 59 | 86 | 37 | 2 | 14 | 1 |
| Total | 7 | 14 | 68 | 134 | 86 | 21 | 268 | 808 |
| | | | | Relative token frequency | | | | |
| *Llevar* | 0.12 | 0.13 | 0.34 | 0.93 | 1.38 | 1.54 | 6.42 | 15.41 |
| *Traer* | 0.71 | 1.66 | 2.24 | 1.67 | 1.07 | 0.16 | 0.35 | 0.02 |
| Total | 0.83 | 1.79 | 2.58 | 2.6 | 2.45 | 1.7 | 6.77 | 15.43 |
| | | | | Absolute type frequency | | | | |
| *Llevar* | 1 | 1 | 6 | 34 | 38 | 18 | 173 | 392 |
| *Traer* | 5 | 8 | 35 | 43 | 26 | 2 | 12 | 1 |
| Total | 6 | 9 | 41 | 77 | 64 | 20 | 185 | 392 |
| | | | | Relative type frequency | | | | |
| *Llevar* | 0.12 | 0.13 | 0.23 | 0.66 | 1.07 | 1.45 | 4.37 | 7.49 |
| *Traer* | 0.59 | 1.02 | 1.33 | 0.83 | 0.73 | 0.16 | 0.30 | 0.02 |
| Total | 0.71 | 1.15 | 1.56 | 1.49 | 1.8 | 1.61 | 4.67 | 7.51 |

If we focus on the productivity of both causative micro-constructions together (see the numbers starting with 'Total'), it is observed that type and token frequencies slowly increase between the 13th and 15th centuries. This can be interpreted as a sign of the development of a subschema of the causative-construction with CAM verbs and its increasing grammaticalization. Between the 15th and 18th centuries, its productivity remained relatively stable, with even a small decrease in the 18th century.[8] From the 19th century on, the productivity of the subschema increased strongly. Thus, the productivity profile of the subschema follows the typical S-shaped curve with a slow beginning followed by a rapid expansion (Nevalainen 2015). Considering this rise in productivity, it is to be expected that the subschema has undergone formal changes over time that reveal an increasing grammaticalization pattern.

Next, when comparing both micro-constructions, the results of the productivity analysis show that the causative micro-construction with *traer* was by far the most productive one in Old Spanish. Its relative type and token frequencies reach its maximum productivity in the 15th century (resp. 2.24 and 1.33), after which it starts to lose productivity until it becomes marginal from the 18th century onwards (e.g., with a relative token frequency of 0.02 in the 20th c.). On the contrary, the micro-construction with *llevar* displays an uninterrupted increase in its type and token frequencies throughout the history of Spanish, surpassing the

micro-construction with *traer* in the 17th century and experiencing a particularly marked increase in productivity in the 19th and 20th centuries.

In the light of these data, we can distinguish three phases in the internal development of the subschema, which coincide with the well-known periodization of Spanish in three stages (Cano-Aguilar 1988): (a) a first phase in which the micro-construction with *traer* was dominant, which corresponds to the Old Spanish period (13th to 15th centuries), (b) a phase of competition between both micro-constructions, during the Classical Spanish period (16th and 17th centuries) and (c) a last phase in which the micro-construction with *llevar* is clearly dominant, which coincides with the Modern Spanish period (18th to 20th centuries). Thus, the main changes in the internal development of the subschema coincide with two moments of major transformations in the linguistic system of Spanish. The beginning of the decrease in productivity of *traer* in favor of *llevar* takes place during the transition between Old and Classical Spanish, while the total decline of the micro-construction with *traer* occurs in the "Early Modern Spanish" period (Octavio de Toledo Huerta 2016), which spans from the end of the 17th century to the beginning of the 19th century.

The predominance of *traer* in Old Spanish and the later development of the micro-construction with *llevar* can be explained by the Aktionsart characteristics of *llevar* and *traer* as motion verbs (see Section 2). We already know that the subschema of the causative construction with CAM verbs has its origin in metaphorical expressions in which an event is conceptualized as the Goal of a displacement. Therefore, for a motion verb to enter into the subschema, it must be able to express displacements towards a syntactically realized Goal. The tendency of *llevar* to express Source-oriented movements without a Goal bounding the event hindered its incorporation into the subschema in Old Spanish. In contrast, the tendency of *traer* to express Goal-oriented motion events facilitated the productivity of the causative micro-construction with *traer* in the early period, compared to *llevar*.

From the 16th century on, the loss of productivity of the micro-construction with *traer* and the expansion of the micro-construction with *llevar* can be explained by the consolidation of the deictic opposition of *llevar* and *traer* as motion verbs, and the subsequent loss of their original aspectual properties. In the 16th century, *traer* has become entirely a coming verb, hence the marked term of the verb pair.[9] Since there is no connection between the deictic center and the caused event, the micro-construction with *traer* became less semantically transparent, and as a consequence, its productivity decreased. On the contrary, once *llevar* lost its tendency to express Source-oriented motion events, it became more suitable to be used in the causative construction. The beginning of the last phase, in which *traer* is residual and *llevar* is clearly predominant (18th to 20th centuries), coincides with a further restriction of the meaning of *traer* as a deictic motion verb, so that it can only express movements towards the place occupied by the speaker.

### *4.2. Formal Features*

The increase in productivity of the potential subschema with CAM verbs, led by the micro-construction with *llevar*, can be an indicator of its ongoing grammaticalization. It is possible, therefore, that the micro-constructions under study, especially the one with *llevar*, have undergone formal changes typical of grammaticalization processes. This subsection examines four formal parameters. The first one concerns the fixation of the micro-constructions based on a variational analysis of the prepositional slot (Section 4.2.1). The three other parameters, namely, the position of causee (Section 4.2.2), the degree of adjacency (Section 4.2.3) and association (Section 4.2.4), measure and compare their degree of unithood.

### 4.2.1. Fixation of the Prepositional Slot

As mentioned in Section 1, the causative micro-constructions with *llevar* and *traer* typically present the following formal structure: [NP1 + *llevar*/*traer* + NP2 + *a* + infinitive]. However, the corpus analysis shows that, in former stages, other structural variants existed without prepositions, as in (9a–b), or with the preposition *en* ('in') instead of *a* ('to') (9c).[10]

(9) a. Deseaba que ya que mi buena suerte *me **había llevado** gozar* de su alegre vista, me concediese lograr el llamarme suya. (Andrés Sanz del Castilllo, *La mojiganga del gusto*, 1641).

'I wished that, since my good fortune had **led** me to enjoy his joyful sight, it would grant me to call me his'.

b. El juego, que por sí mismo y de su naturaleza, ***trae** recrear* el ánimo. (Francisco de Luque Fajardo, *Fiel desengaño contra la ociosidad y los juegos*, 1603).

'Gambling, which by itself and by its nature, **leads** to entertain the spirit'.

c. La composición del hombre *nos deve **traer** en alabar y servir y amar* al artífice, que es Dios. (Pedro Mejía, *Silva de varia lección*, 1540–1550).

'The composition of the human being should **lead** us to praise and serve and love the artificer, who is God'.

Since the preposition is a fixed component of the construction, the reduction in the paradigmatic variability of this slot could be interpreted as an indication of the grammaticalization of the pattern (Lehmann 2015). Therefore, all the occurrences in the CDH responding to the following formal structures were collected: [NP1 + *llevar*/*traer* + NP2 + *en* + infinitive] and [NP1 + *llevar*/*traer* + NP2 + infinitive].[11] Table 2 presents a historical overview of the absolute frequencies of the micro-constructions under study and the variants without prepositions and with the preposition *en*.

**Table 2.** Constructional variants with the preposition *en* and without prepositions.

|  | **13** | **14** | **15** | **16** | **17** | **18** | **19** | **20** |
|---|---|---|---|---|---|---|---|---|
| *Llevar* + <a + INF> | 1 | 1 | 9 | 48 | 49 | 19 | 254 | 807 |
| *Llevar* + <Ø + INF> | - | - | 1 | - | 1 | - | - | - |
| *Traer* + <a + INF> | 6 | 13 | 59 | 86 | 37 | 2 | 14 | 1 |
| *Traer* + <en + INF> | - | 1 | 1 | 1 | - | - | - | - |
| *Traer* + <Ø + INF> | - | 2 | 14 | 1 | 1 | - | - | - |

The results of the Table 2 show that micro-constructions with the preposition *a* were most frequent in all periods, and that the variant with the preposition *en* has always been exceptional. The causative micro-construction with *traer* showed some paradigmatic variation between the preposition *a* and absence of a preposition, especially in the 15th century. On the contrary, the causative micro-construction with *llevar* has presented a higher degree of formal fixation all over time, as only two occurrences without prepositions could be retrieved. This indicates that the micro-construction with *llevar* presents lower paradigmatic variability than the one with *traer*. However, from the 18th century onwards, the whole subschema became fixed with the preposition *a*, what is a sign of its grammaticalization (Lehmann 2015).

### 4.2.2. Position of the Causee

In grammaticalization processes, the syntagmatic linkage between the elements of a construction is gradually reinforced, which is known as syntactic incorporation (Correia Saavedra 2021; Lehmann 2015). In verbal grammaticalization, it is thus expected that the presence of lexical items interposed between the auxiliary verb and the non-finite verb will become less frequent over time (Heine 1993; Rodríguez Molina 2010; Torres Cacoullos 2000). Regarding the causative construction, the formal variable that has probably most attracted the attention of scholars is the position of the causee (NP2) when lexically expressed (Enghels and Comer 2020; Enghels and Roegiest 2014; Shibatani and Pardeshi 2002; Da Silva 2004). More concretely, the placement of the causee between the causative verb and the infinitive, which has been called VOV structure, implies a lower degree of incorporation, and thus grammaticalization, between the two verbs (10).

(10) a. El despecho ***llevaba*** *a Pablo* *a hacer* alarde de una indiferencia despreciativa (Fernán Caballero. (Cecilia Böhl de Faber), *Clemencia*, 1852).

'The spite **led** Pablo to display a disdainful indifference'.

b. Onde devedes saver que quando el peccado non es purgado por penitençia, por la su grand pesadunbre ***trahe*** *al peccador* *a caher* en otros. (*Un sermonario castellano medieval*, 15th century).

'Therefore, you must know that when the sin is not purged by penance, for its big weight it **leads** the sinner to fall into others'

On the contrary, the postposition of the causee, which has been called VV structure, is interpreted as a sign of more incorporation and unithood, hence a more advanced stage of grammaticalization of the construction (11).

(11) a. Las observaciones anteriores *han **llevado*** *a afirmar* *a los escrituristas* que el pueblo hebreo y, por lo tanto, la Biblia, tiene un cierto sentido lineal del tiempo. (Luis Maldonado, *La plegaria eucarística. Estudio de teología bíblica y litúrgica sobre la misa*, 1967).

'The above observations have **led** writers to assert that the Hebrew people, and therefore the Bible, have a certain linear sense of time'.

b. el demonio tienta experimentalmente si podrá ***traer*** *a pecar* *a los hombres*. (Juan de Pineda, *Diálogos familiares de la agricultura cristiana*, 1589).

'The Devil tries experimentally if he will be able to **lead** people to commit sins'.

According to Da Silva (2004), the two syntactic schemes—VOV and VV—profile different elements of the event: the VOV scheme profiles the causee, presented as a more autonomous entity, while the VV scheme profiles the caused event as a whole. Thus, the VOV structure is related to indirect causation events, in which the causee acts autonomously, while the VV structure is linked to direct causation, so the causer has more control over the caused event (Section 4.3.1 further details the concepts of direct and indirect causation). Considering that the micro-constructions under study have been linked to indirect causation, the VOV structure is expected to predominate. However, over the centuries, further grammaticalization of the construction would entail an increasing number of examples with the VV structure. Table 3 presents quantitative data on the position of the nominal causee per century.[12]

**Table 3.** Position of the NP2 in the micro-constructions with *llevar* and *traer*.

| | 13 | 14 | 15 | 16 | 17 | 18 | 19 | 20 |
|---|---|---|---|---|---|---|---|---|
| *Llevar* VOV | - | - | 3 (100%) | 5 (100%) | 1 (33%) | 2 (100%) | 23 (96%) | 111 (94%) |
| *Llevar* VV | - | - | - | - | 2 (67%) | - | 1 (4%) | 7 (6%) |
| Total *llevar* | - | - | 3 | 5 | 3 | 2 | 24 | 118 |
| Traer VOV | 1 (100%) | 3 (100%) | 11 (92%) | 8 (80%) | 3 (75%) | - | 2 (100%) | - |
| *Traer* VV | - | - | 1 (8%) | 2 (20%) | 1 (25%) | - | - | - |
| Total *traer* | 1 | 3 | 12 | 10 | 4 | - | 2 | - |

Table 3 shows that the causative micro-constructions with *llevar* and with *traer* are preferably expressed through the VOV structure in all epochs. The only exception is the micro-construction with *llevar* in the 17th century, although very few examples are examined.

The overall image points to a constant low degree of incorporation and unithood in both CAM-based micro-constructions. This is quite remarkable given that other causative subschemas show a diachronic tendency towards increasing postposed causees. For example, the proportion of examples with the postposed causees in the micro-construction with *hacer* increases from 50% in Old Spanish to 92% in Modern Spanish (Davies 1995). Similarly, in the micro-construction with *poner*, the postposition of the causee represents 25% of examples in the 13th century and ascends to 60.8% in the 21st century (Enghels and Comer 2020). This suggests that the potential subschema with CAM verb is a rather fixed pattern with very little syntactic variation across time.

### 4.2.3. Position of Adjuncts

In addition to the causee, adjunct complements (including adverbs and prepositional phrases) allow high mobility in the construction. Consequently, their position can also shed light on the degree of incorporation between the causative verb and the infinitive, and hence on the degree of unithood of the construction (Enghels and Comer 2018; Rodríguez Molina 2010; Torres Cacoullos 2000). Adjuncts can be placed between the causative verb and the infinitive (12), or before or after these constituents (13).

(12) a. No será éste un método que ***lleve*** *simplemente a resolver* los problemas que las cosas plantean. (Xavier Zubiri, *Naturaleza, Historia, Dios*, 1932–1944).

'This will not be a method that **leads** simply to solving the problems that things pose'.

b. La congoja, y deseo la **traía** muchas veces á desfallecer y desmayarse. (Fray Luis de Léon, Exposición del Cantar de los Cantares, 1561).

'The distress and desire **led** her many times to give out and faint'.

(13) a. Lo que acabamos de decir nos ***lleva*** *a tratar* ahora de la constitución general de nuestro globo. (S. Alvarado, *Ciencias Naturales*, 1957–1974).

'What we have just said leads us to address now the general constitution of our globe'.

b. Suplica a Dios omnipotente [ ... ], que por su alto juicio le ***ha traído*** *a suceder* en esta monarquía. (Luis Cabrera de Córdoba, *Historia de Felipe II, rey de España*, 1619).

'He begs almighty God [ ... ], which by his high judgement has **led** him to succeed in this monarchy'.

Thus, it is expected that a higher degree of grammaticalization of the micro-constructions with *llevar* and *traer* coincides with less adjuncts to be found between the causative verb and the infinitive. Table 4 shows the quantitative results of interpolation of adjuncts.[13]

**Table 4.** Interpolation of adjuncts between the causative verb and the infinitive.

|  | 13 | 14 | 15 | 16 | 17 | 18 | 19 | 20 |
|---|---|---|---|---|---|---|---|---|
| *Llevar* interpolation | - | - | 2 (67%) | 4 (27%) | 4 (27%) | 4 (36%) | 26 (36%) | 70 (45%) |
| *Llevar* non-interpolation | - | - | 1 (33%) | 11 (73%) | 11 (73%) | 7 (64%) | 46 (64%) | 87 (55%) |
| Total *llevar* | - | - | 3 | 15 | 15 | 11 | 72 | 157 |
| *Traer* interpolation | - | 1 (50%) | 1 (8%) | 5 (28%) | 3 (25%) | - | 1 (17%) | - |
| *Traer* non-interpolation | - | 1 (50%) | 11 (92%) | 13 (72%) | 9 (75%) | - | 5 (83%) | - |
| Total *traer* | - | 2 | 12 | 18 | 12 | - | 6 | - |

The data show that, unlike the causee (NP2),[14] adjuncts are most frequently anteposed to the auxiliary verb or postposed to the infinitive, rather than interposed between the causative verb and the infinitive in both micro-constructions. Except for the 14th century, where there are only two examples with adjuncts, the micro-construction with *traer* shows a clear predominance of the pattern without interpolation and a relatively low proportion of intercalated adjuncts, ranging between 8% and 28%. Regarding the micro-construction with *llevar*, the pattern without interpolation also predominates, but to a lesser extent, since the number of interposed adjuncts ranges between 27% and 45%. An exception is the 15th century, in which the higher proportion of examples with interpolation is due to the small number of relevant cases observed.

However, a closer look at the data shows that the proportion of interposed adjuncts is very similar in both constructions in the 16th and 17th centuries (between 25% and 28%), but after the 18th century, it increases in the micro-construction with *llevar*, reaching its peak in the 20th century (45%). This suggests that, after a period in which both micro-constructions showed a similar degree of syntactic incorporation, unexpectedly, the micro-construction with *llevar* gradually became less incorporated. From a more general perspective, the data

show that the potential subschema with CAM verbs does not seem to reach a higher degree of syntactic incorporation over time.

### 4.2.4. Degree of Association

The last formal parameter focuses on the combination of the causative verb with more than one (14) or one (15) infinitive.

(14) a. Una poderosa inclinacion *me **lleva** á respetarlo, y á prestarle* en todas ocasiones testimonios de veneracion. (Fray Francisco Alvarado, *Cartas críticas del Filósofo Rancio*, 1811).

'A powerful inclination **leads** me to respect him, and to render him on all occasions testimonies of veneration'.

b. Este abominable vicio que casi del todo ***trae** a aborrescer y menospreciar* a Dios. (Jaime Montañés, *Espejo de bien vivir y para ayudar a bien morir*, 1573–1577).

'This abominable vice that almost entirely **leads** one to hate and despise God'.

(15) a. El mismo estado de debilidad en que se siente, y que *le **lleva** á buscar* la fuerza de que carece como individuo aislado. (Casildo Ascárate y Fernández, *Insectos y criptógamas que invaden los cultivos en España*, 1893).

'The same state of weakness in which he feels, and that leads him to seek the strength that he lacks as an isolated individual'.

b. La fortuna, [ . . . ], me **ha traído** a referir adversidades. (Francisco Manuel de Melo, Historia de los movimientos, separación y guerra de Cataluña, 1645).

'Fortune, [ . . . ], has **brought** me to report adversities.'

This parameter is based on the premise that the association of an auxiliary with a single non-finite verb is a sign of a higher degree of unithood of the construction and, consequently, more advanced grammaticalization. On the contrary, the combination of an auxiliary with several non-finite verbs indicates that the auxiliary has a broader structural scope, which is considered as a potential sign of a lower degree of grammaticalization (Comer 2020; Torres Cacoullos 2000). Based on this parameter, Comer (2020) observes that, in present-day Spanish, the causative micro-construction with *meter* combines more frequently with multiple infinitives than the more grammaticalized variant with *poner*. If the causative micro-constructions with *llevar* and *traer* are undergoing a grammaticalization process, a decrease in the frequency of examples with multiple infinitives is expected. Table 5 shows the results.

**Table 5.** Degree of association between the causative verb and the infinitive.

|  | **13** | **14** | **15** | **16** | **17** | **18** | **19** | **20** |
|---|---|---|---|---|---|---|---|---|
| *Llevar* one INF | 1 (100%) | 1 (100%) | 9 (100%) | 43 (90%) | 47 (96%) | 14 (74%) | 229 (90%) | 749 (93%) |
| *Llevar* multiple INFs | - | - | - | 5 (10%) | 2 (4%) | 5 (26%) | 25 (10%) | 58 (7%) |
| Total *llevar* | 1 | 1 | 9 | 48 | 49 | 19 | 254 | 807 |
| *Traer* one INF | 6 (100%) | 13 (100%) | 56 (95%) | 78 (91%) | 34 (92%) | 2 (100%) | 13 (93%) | 1 (100%) |
| *Traer* multiple INFs | - | - | 3 (5%) | 8 (9%) | 3 (8%) | - | 1 (7%) | - |
| Total *traer* | 6 | 13 | 59 | 86 | 37 | 2 | 14 | 1 |

The data show that both micro-constructions are mostly combined with a single infinitive. In the case of *llevar*, between the 13th and 15th centuries, no cases with multiple infinitives are documented, which may be due to the low number of tokens overall. Between the 16th and 20th centuries, the proportion of examples with multiple infinitives ranges between 4% and 10%, except for the 18th century, what may also be partially explained by the relative data scarcity (see note 7). As far as *traer* is concerned, cases with multiple infinitives are documented with a frequency that ranges between 5% and 9% between the 15th and 17th centuries and in the 19th century. In the other centuries, no examples with

multiple infinitives are found, due to the low number of occurrences. These data suggest that the micro-constructions under study present a comparable and relatively stable degree of association between the causative verb and the infinitive.

To recap, the analysis of a set of formal features has shown that both CAM micro-constructions have evolved in a rather similar way. Regarding its degree of grammaticalization, the causative subschema with CAM verbs is characterized by a first stage of variation in the prepositional slot followed by the fixation of the preposition *a* from the 18th century on. However, the three remaining parameters indicate that there is no increase in the degree of unithood of the construction over time, hence no further grammaticalization. More concretely, there is (a) a constant predominance of the VOV pattern, (b) a relatively stable degree of association between the causative verb and the infinitive and (c) a growing proportion of interposed adjuncts suggesting even a decreasing degree of syntactic incorporation.

The observation that the subschema has not evolved towards greater unithood over time could be due to a hypothetical semantic specialization in the expression of indirect causation (see Section 4.3.1). Constructions that specialize in indirect causation are typically associated with the VOV pattern (Da Silva 2004) and have a lower degree of grammaticalization than those expressing direct causation events (Shibatani and Pardeshi 2002). In the following section, the functional analysis of the micro-constructions will reveal that the lack of an increase in the unithood of the subschema is indeed motivated by semantic factors.

*4.3. Semantic Features*

This section presents the results of a functional analysis of the causative micro-constructions with *llevar* and *traer* based on two parameters: the animacy of the participants (Section 4.3.1) and the semantics of the infinitive (Section 4.3.2). It explores whether the variants expand to new semantic domains over time, what is expected when a more abstract subschema develops or whether they have specialized in the expression of a specific type of causative event.

4.3.1. Animacy of the Causer and the Causee

The most widespread semantic classification of causative events distinguishes between direct and indirect causation. In an event of direct causation, the causer physically manipulates the causee to perform an action, so that the causee has no autonomy or control over the caused event, but the causer does. In a process of indirect causation, on the contrary, the causer exerts a more restricted influence on the causee and, as a result, the causee autonomously performs an action, which may take place at a different time and place (Shibatani and Pardeshi 2002; Lavale 2013). According to Enghels and Comer (2020), prototypical direct causation has an animate causer and an inanimate causee, as in example (16a), whereas prototypical indirect causation involves an inanimate causer and an animate causee, as in (16b).

(16) a. Bien asy como <u>el libre escriuano</u> **trae** <u>la pluma</u> a escreuir las razones que el quiere. (Mose Arragel de Guadalfajara, *Traducción y glosas de la Biblia de Alba*, 1422–1433).

'Right so as <u>the free scribe</u> **makes** <u>the quill</u> write the reasons that he wants'.

b. *Mi triunfo primero me **llevó** á buscarlos* contínuos, y á conseguirlos tambien (Antonio Alcalá Galiano, *Memorias*, 1847–1849).

'<u>My first triumph</u> **led** <u>me</u> to look for them continuously, and to get them too'.

In examples such as (16a), the causer (in this case, *el libre escriuano* 'the free scribe') is an animate, agentive being that physically manipulates the causee (*la pluma* 'the quill'), an inanimate entity lacking all control and autonomy. Consequently, the caused event (*escribir* 'write') necessarily occurs simultaneously with the causing event (the action of the scribe on the quill). In contrast, in (16b), the causer is an inanimate entity (in this case, a dynamic state of affairs, *mi triunfo primero* 'my first triumph') that influences the causee (*me* 'me') in such a way that the latter, an animate and agentive being, gets involved in another action.

Thus, the causing event does not necessarily occur in the same moment or place as the caused event. In fact, in (14b), it is inferred that the causing event (the first triumph) occurs first and that, subsequently, the caused event is carried out (*buscarlos continuos [los triunfos]* 'look for them continuously [the triumphs]'). Thus, the examples with animate causer and inanimate causee, as well as the examples with inanimate causer and animate causee, clearly correspond to the categories of direct causation and indirect causation, respectively, and represent prototypical cases of each type. Other types of events may lie at intermediate points on the continuum between these two prototypes. Several attempts have been made to characterize and classify the intermediate cases between prototypical direct and indirect causation (e.g., Comer 2020, pp. 486–91; Shibatani and Pardeshi 2002). Besides the animacy of the participants, other factors, such as dynamicity or control, are also involved (Enghels and Comer 2020). For instance, the combination of two animate participants mostly results in a situation where the causer induces the causee to perform an action by psychological means (Guilquin 2010), as in example (17a). However, there are also some cases in which the causer acts physically on the causee to force him or her do something, as in example (17b), extracted from Comer (2020, p. 488).

(17) a. <u>Alcina</u>, de quien dize Orlando, que por engaño **traya** <u>los hombres</u> a gozar de sus regalos. (Juan de Luna, Diálogos familiares en lengua española, 1619).

> '<u>Alcina</u>, of whom Orlando says that she deceitfully **led** <u>men</u> to enjoy her gifts'.

b. La levanta del asiento, engancha sus manos entre las carnes mórbidas de su brazo y tira de ella hasta ***ponerla*** <u>*a caminar*</u>. (Sánchez-Andrade, Cristina, *Bueyes y rosas dormían*, 2001).

> 'He lifts her from the seat, hooks her hands between the morbid flesh of his arm and pulls her up until <u>*he*</u> ***makes*** <u>*her walk*</u>'.

In (17a), the causer (*Alcina*) induces the causee (*los hombres* 'men') to get involved in another event (*gozar de sus regalos* 'enjoy her gifts') by verbal and psychological means, namely *por engaño* ('deceitfully'). As the causee performs the caused event autonomously, likely at a different time than the causing event, a situation like the one in (17a) can be categorized as indirect causation. On the contrary, in (17b), the causer exerts a physical force on the causee (*la* 'her'), who, despite being animate, lacks autonomy and control over the caused event (*caminar* 'walk'), which occurs (almost) simultaneously with the causing event. Therefore, examples like (17b) should be categorized as direct causation. A qualitative analysis of our corpus reveals that, when causative micro-constructions with *llevar* and *traer* feature two animate participants, causation is always achieved by verbal, social or psychological means, as in (17a), while situations of physical causation such as the one in (17b) are not documented. Therefore, all examples in our corpus with two animated participants are considered cases of indirect causation.

Another possibility is that both the causer and the causee are inanimate entities, as in (18). In these cases, the causer exerts a physical force on the causee. The situation results in the caused event, without the causer nor the causee having any control over it. Since the causee does not act autonomously, and the caused event is simultaneous to the causing event, the situation described is closer to direct causation and is categorized as such.

(18) <u>El mismo mecanismo de rotación o enrollamiento</u> **lleva** el fórnix gástrico a colo-carse en contacto con el tercio inferior del esófago. (M. Díaz Rubio, Lecciones de patología y clínica médica. Aparato digestivo, 1964).

> 'The same mechanism of rotation or coiling **leads** the gastric fornix to stand into contact with the lower third of the esophagus'.

It follows from this that the animacy of the participants is an important parameter to characterize the type of causation expressed by a construction. As mentioned before (supra Section 1), the causative micro-construction with *llevar* has been placed within the scope of indirect causation (Da Silva 2004, 2012). Still, it has been noted that the emergence of a new grammatical construction usually coincides with an expansion of the construction into new,

less prototypical semantic types, as a consequence of the abstraction of the schema (Barðdal 2008; Traugott and Trousdale 2013). If we assume, following Da Silva (2004, 2012), that the micro-constructions with *llevar* and *traer* express prototypically indirect causation, it is reasonable to think that, in Old Spanish, they were already strongly associated with events of indirect causation, and that over time they expressed more diverse types of causation and more flexibly admitted different types of participants.

With regard to the nature of the causer, if the formation of the micro-constructions under study has led to the development of a new causative subschema, it is expected that they initially selected inanimate and later more easily accepted animate causers (NP1). Table 6 presents the results of this classification.[15]

**Table 6.** Animacy of the causer (NP1).

|  | 13 | 14 | 15 | 16 | 17 | 18 | 19 | 20 |
|---|---|---|---|---|---|---|---|---|
| *Llevar* NP1 [anim] | - | 1 (100%) | 1 (11%) | 5 (12%) | 8 (17%) | 2 (12%) | 11 (4%) | 27 (3%) |
| *Llevar* NP1 [inanim] | - | - | 8 (89%) | 38 (88%) | 40 (83%) | 14 (88%) | 242 (96%) | 748 (97%) |
| Total *llevar* | - | 1 | 9 | 43 | 48 | 16 | 253 | 775 |
| *Traer* NP1 [anim] | 3 (60%) | 5 (38%) | 20 (39%) | 34 (43%) | 10 (27%) | - | 1 (8%) | - |
| *Traer* NP1 [inanim] | 2 (40%) | 8 (62%) | 31 (61%) | 45 (57%) | 27 (73%) | 2 (100%) | 12 (92%) | 1 (100%) |
| Total *traer* | 5 | 13 | 51 | 79 | 37 | 2 | 13 | 1 |

The results show that both micro-constructions select mostly inanimate causers, although not entirely to the same degree. The micro-construction with *traer* appears with animate causers with some frequency until the 17th century. For instance, in the 16th century, cases with animate causers account for 43% of the total number of cases. However, from the 17th century on, the number of animate causers decreases, coinciding with the decline in productivity of the micro-construction. With respect to the micro-construction with *llevar*, the proportion of animate causers is much lower, ranging between 11% and 17% between the 15th and 18th centuries and decreasing drastically in the 19th and 20th centuries. These data suggest that, rather than an expansion of the construction to new semantic domains, there was a specialization of the micro-constructions under study towards prototypical indirect causation, since they select inanimate causers with increasing rigidity.

Since prototypical indirect causation involves an animate causee (Enghels and Comer 2020), it is expected that the causative micro-constructions with *llevar* and *traer* initially selected animate causees and that, over time, there was an increase in inanimate ones. The semantic classification of the causee (NP2) yields the results presented in Table 7.

**Table 7.** Animacy of the causee (NP2).

|  | 13 | 14 | 15 | 16 | 17 | 18 | 19 | 20 |
|---|---|---|---|---|---|---|---|---|
| *Llevar* NP2 [anim] | - | 1 (100%) | 6 (67%) | 43 (90%) | 44 (90%) | 16 (84%) | 245 (96%) | 790 (98%) |
| *Llevar* NP2 [inanim] | 1 (100%) | - | 3 (33%) | 5 (10%) | 5 (10%) | 3 (16%) | 9 (4%) | 17 (2%) |
| Total *llevar* | 1 | 1 | 9 | 48 | 49 | 19 | 254 | 807 |
| *Traer* NP2 [anim] | 5 (83%) | 11 (85%) | 56 (95%) | 82 (95%) | 35 (95%) | 2 (100%) | 14 (100%) | 1 (100%) |
| *Traer* NP2 [inanim] | 1 (17%) | 2 (15%) | 3 (5%) | 4 (5%) | 2 (5%) | - | - | - |
| Total *traer* | 6 | 13 | 59 | 86 | 37 | 2 | 14 | 1 |

The results show that, from the beginning, both micro-constructions were mostly formed with animate causees. Moreover, in both cases, there is a clear trend towards an increasingly rigid selection of animate causees. In concrete, in the micro-construction with *traer*, the proportion of inanimate causees is reduced from 17% (13th century) to 0% (18th to 20th centuries), while in the case of *llevar*, it is reduced from 33% (15th century) to 2% (20th century). These data reinforce the idea that, rather than expanding their scope, the

micro-constructions under study became increasingly specialized in order to express events of indirect causation.

Finally, to get a clearer picture of the semantic profile of the micro-constructions with *llevar* and *traer* as a whole, the examples are classified according to the animacy of both participants (NP1 and NP2) in Table 8.

**Table 8.** Animacy of the causer (NP1) and the causee (NP2).

|  | 13 | 14 | 15 | 16 | 17 | 18 | 19 | 20 |
|---|---|---|---|---|---|---|---|---|
| *Llevar* NP1 [anim] + NP2 [inanim] | - | - | 1 (11%) | - | 2 (4%) | - | 6 (2%) | 7 (1%) |
| *Llevar* NP1 [inanim] + NP2 [inanim] | - | - | 2 (22%) | 4 (9%) | 3 (6%) | 1 (6%) | 3 (1%) | 10 (1%) |
| *Llevar* NP1 [anim] + NP2 [anim] | - | 1 (100%) | - | 5 (12%) | 6 (13%) | 2 (13%) | 5 (2%) | 20 (3%) |
| *Llevar* NP1 [inanim] + NP2 [anim] | - | - | 6 (67%) | 34 (79%) | 37 (77%) | 13 (81%) | 239 (94%) | 758 (95%) |
| Total *llevar* | - | 1 | 9 | 43 | 48 | 16 | 253 | 795 |
| *Traer* NP1 [anim] + NP2 [inanim] | - | - | 1 (2%) | 1 (1%) | 1 (3%) | - | - | - |
| *Traer* NP1 [inanim] + NP2 [inanim] | - | 2 (15%) | 2 (4%) | 1 (1%) | 1 (3%) | - | - | - |
| *Traer* NP1 [anim] + NP2 [anim] | 3 (60%) | 5 (38%) | 19 (37%) | 33 (42%) | 9 (24%) | - | 1 (8%) | - |
| *Traer* NP1 [inanim] + NP2 [anim] | 2 (40%) | 6 (46%) | 29 (57%) | 44 (56%) | 26 (70%) | 2 (100%) | 12 (92%) | 1 (100%) |
| Total *traer* | 5 | 13 | 51 | 79 | 37 | 2 | 13 | 1 |

The results presented in Table 8 confirm the observed tendency towards semantic specialization. Up until the 16th century, the micro-construction with *traer* is easily employed to express situations with an inanimate causer and an animate causee, as in (19a), and also with two animate participants (19b). The frequency of examples with both animate participants ranges from 60% to 37% until the 16th century. From the 17th century on, there is a drop with no return.

(19) a. La natura spiral **trae** <u>al omne</u> a saber todas las cosas segunt son. (Libro de los buenos proverbios que dijeron los filósofos y sabios antiguos, 1250).

'The spiritual nature **leads** <u>men</u> to know all things as they are'.

b. <u>Traidor</u>, que con tu ley halaguera <u>me</u> engañaste, y **has traído** *a dexar* la verdadera. (Jorge Manrique, *Los fuegos*, 1469–1479).

'<u>Traitor</u>, who with your flattering law have deceived me, and have **brought** <u>me</u> to forsake the true one'.

In contrast, the micro-construction with *llevar* with two animate participants (20) was always rare, with a proportion ranging between 12% and 13% between the 16th and 18th centuries. In the 19th and 20th centuries, the frequency of this type declines further to 2% and 3%.

(20) Algun dulce <u>el maestro al niño</u> dando, lo **lleva** *así á estudiar* más fácilmente. (Manuel María Arjona, *Poesías*, 1790–1820).

'The teacher gives some candy to the child, so he **leads** him to study more easily.'

With both micro-constructions, the possibility of expressing causative events with both inanimate participants, as shown supra in (18), or with an animate causer and an inanimate causee, as shown supra in (16a), which are types of events related to direct causation, was always infrequent. However, the proportion of such cases drops even further in the 19th and 20th centuries (0–2%).

These data show that the causative micro-constructions with *llevar* and *traer* specialized in events of indirect causation from the earliest documentations on. In this, the micro-constructions with *llevar* and *traer* differ from others, such as the variants with *hacer* and *poner*, which more readily allow the expression of both direct and indirect causation (Enghels and Comer 2020). What is more, far from diversifying their semantic profile, the micro-constructions with *llevar* and *traer* increasingly specialized towards the prototype of indirect causation, which involves an inanimate causer and an animate causee. This type of examples raised from 40% to 100% for *traer* and from 67% to 95% for *llevar*. This suggests that a subschema with CAM verbs developed within the causative construction

with a specialized function, namely the expression of indirect causation. Since indirect causation is associated with less grammaticalized structures (Shibatani and Pardeshi 2002), the further specialization of the subschema in indirect causation events explains why there has been no increase over time of its degree of unithood (see Section 4.2).

However, the ease with which the *traer* micro-construction accepted animated causers up to the 16th century represents an important difference respect to the micro-construction with *llevar*, which from the beginning was used in a more restricted way for prototypical indirect causation. This points to the fact that the semantic profile of the micro-construction with *traer* was progressively attracted to that of the micro-construction with *llevar* in Classical Spanish, a period in which *llevar* started to lead the subschema. The leadership of the micro-construction with *llevar* is explained by the semantic changes experienced by *llevar* and *traer* as motion verbs, which made the meaning of the micro-construction with *llevar* more transparent, and ultimately caused the decline in the micro-construction with *traer*.

### 4.3.2. Semantics of the Caused Event

If speakers abstracted the meaning of the causative micro-constructions with *llevar* and *traer* and created a new subschema, it could be expected that, over time, it would extend to new semantic contexts (Barðdal 2008; Traugott and Trousdale 2013). Moreover, the removal of the original semantic constraints is associated with the grammaticalization of the construction, as a consequence of the loss of its semantic integrity (Lehmann 2015).

In order to approach the lexical-semantic fields with which the micro-constructions under study are associated, the infinitives in the dataset are labelled on the basis of the ADESSE classification (García-Miguel et al. 2010). This classification distinguishes six verbal macro-categories that are further subdivided into more specific categories, up to three levels of analysis. For example, a macro-class of mental verbs is distinguished, which at a second level of analysis is divided into verbs of feeling (e.g., *gustar* 'to like'), verbs of perception (e.g., *ver* 'to see') and verbs of cognition (e.g., *pensar* 'to think'). On their turn, some verbs of cognition are ascribed, at a third level of analysis, to the more specific classes of verbs of knowledge (e.g., *saber* 'to know') and verbs of belief (e.g., *creer* 'to believe'). In order to achieve a precise level of description, the most detailed level of analysis is adopted. Table 9 shows the number of different semantic classes presented by the infinitives in each century.

**Table 9.** Number of different semantic classes of the infinitive per century.

|  | 13 | 14 | 15 | 16 | 17 | 18 | 19 | 20 |
|---|---|---|---|---|---|---|---|---|
| *Llevar* | 1 | 1 | 6 | 20 | 21 | 12 | 39 | 49 |
| *Traer* | 5 | 8 | 21 | 24 | 15 | 2 | 11 | 1 |

These data confirm that, after their emergence, the causative micro-constructions with *llevar* and *traer* became associated with infinitives of increasingly varied meanings. In the case of *llevar*, this trend extends from the 13th century to the 20th century, except for a decline in the 18th century, which may be due to the smaller number of examples. On the other hand, with respect to the micro-construction with *traer*, from the 17th century onwards, the semantic diversity of infinitives is reduced, coinciding with the general decline in the productivity of the micro-construction. The progressive diversification of the semantic classes of the infinitives points to the fact that a constructional schema is developing over time.

In addition, it is worth considering (a) which are the semantic classes expressed by the infinitives, (b) whether the semantic classes have changed over time and (c) whether they are different in the two micro-constructions under study. As was mentioned before, the micro-constructions derive from expressions that typically indicate a metaphorical

movement towards a state conceptualized as an abstract place, as in example (7a), which is now recovered as (21) (see Section 2).

(21) El uno de tus amigos es aquel que te tuelle del mal y te **lieva** <u>al bien</u> (*Libro de los buenos proverbios que dijeron los filósofos y sabios antiguos*, 1250)

'Your friend is the one who takes you out of evil and **leads** you <u>to good</u>'. Therefore, they are expected to be initially associated with attributive and other stative events, and to later include other types of events and actions. Table 10 shows the most frequent semantic classes of the infinitive per century and per micro-construction.[16]

**Table 10.** Most frequent semantic classes of the infinitive per century.

|  | *llevar* | | *traer* | |
|---|---|---|---|---|
|  | **Class INF** | **#** | **Class INF** | **#** |
| 13 |  |  | knowledge | 2 |
| 14 |  |  | attribution | 3 |
|  |  |  | light verbs | 3 |
|  |  |  | knowledge | 2 |
| 15 | knowledge | 3 | light verbs | 9 |
|  | displacement | 2 | communication | 5 |
|  |  |  | life | 5 |
|  |  |  | attribution | 4 |
|  |  |  | permission | 4 |
| 16 | perception | 10 | attribution | 14 |
|  | feeling | 5 | light verbs | 12 |
|  | attribution | 4 | feeling | 8 |
|  | communication | 4 | life | 8 |
|  | knowledge | 3 | competition | 5 |
| 17 | perception | 12 | attribution | 6 |
|  | communication | 4 | perception | 5 |
|  | life | 4 | feeling | 5 |
|  | attribution | 3 | life | 5 |
|  | possession | 3 | activity | 3 |
| 18 | attribution | 3 |  |  |
|  | communication | 3 |  |  |
|  | perception | 3 |  |  |
|  | feeling | 2 |  |  |
| 19 | perception | 25 | attribution | 2 |
|  | feeling | 21 | communication | 2 |
|  | communication | 19 | light verbs | 2 |
|  | cognition | 16 |  |  |
|  | belief | 13 |  |  |
| 20 | communication | 110 |  |  |
|  | perception | 73 |  |  |
|  | cognition | 55 |  |  |
|  | belief | 51 |  |  |
|  | knowledge | 49 |  |  |

Table 10 reveals important differences between the micro-constructions and between different language stages. From its first attestations, the micro-construction with *llevar* is frequently associated with mental verbs, especially with the subclasses of perception verbs, such as *buscar* ('search') (22a), and verbs of cognition, such as *pensar* ('think') (22b), which is the most frequent infinitive in the 19th and 20th centuries. Moreover, from the 16th century

on, and especially in the 20th century, it is often employed for communication events, as with the verb *expresar* ('express') (20c).

(22) a. Hay en mi corazón una atracción secreta, que *me **lleva** buscaros*. (Emilio Castelar, *Ernesto: novela original de costumbres*, 1855).

'There is in my heart a secret attraction, which **leads** me to look for you'.

b. Estas cosas *le **llevan** a uno a pensar* en la vida. (Miguel Delibes, *Diario de un emigrante*, 1958).

'These things **lead** one to think about life'.

c. El sentimiento religioso ***lleva** al hombre a expresar* su retorno hacia Dios. (Rafael Alcocer, *Iniciación litúrgica: la misa*, 1935).

'Religious sentiment **leads** men to express their return to God'.

Regarding the micro-construction with *traer*, unlike *llevar*, it is very often associated with light verbs, particularly with *hacer* ('make') in examples such as (21), in all epochs except the 17th century.

(23) No diga el que ahorcan que su hado lo traxo a morir aquella muerte, [ . . . ], que lo que *los **trae** a hazer tan ruyn fin* de su vida es su poca consideración. (Antonio de Torquemada, *Jardín de flores curiosas*, 1569).

'Do not say that the one who is hanged that his fate brought him to die that death, [ . . . ], that what **leads** them to make such a dastardly end to their life is their lack of consideration'.

Moreover, in their early stages, both micro-constructions frequently expressed events of attribution, particularly with the verb *ser* 'to be' (24). However, while in the case of *traer* this association is much more frequent and remains constant throughout history (14th to 19th centuries), in the case of *llevar*, it weakened after the 18th century.

(24) a. Que enbidia nin cobdicia de plata nin dineros non les busquen nin ***trayan** a ser* fallesçederos. (Pero López de Ayala, *Rimado de Palacio*, 1378–1406).

'That envy nor greed for silver or money do not seek them nor **lead** them to be perishable'.

b. No he tenido yo la culpa, sino quien *le **ha llevado** de la mano a ser* tan loco. (Lope de Vega, *La prudente venganza*, 1623).

'It was not my fault, but of the one who **led** him by the hand to be so crazy'.

This seems to indicate that both micro-constructions were at some point linked to attributive caused events, as it was expected because of their origin, although to different extents. Over time, both expanded to different semantic domains, but the variant with *traer* remained more closely linked to stative event types. This suggests a lower degree of grammaticalization of the micro-construction with *traer*, which does not experience a loss of its semantic integrity as markedly as *llevar*.

## 5. Discussion and Conclusions

This paper has studied the diachrony of causative micro-constructions with the verbs *llevar* ('take') and *traer* ('bring'). The analysis of seven grammaticalization parameters has shown that the two micro-constructions share a series of formal and semantic features that differentiate them from other causative micro-constructions. As an outcome of the analysis, we propose that a specific subschema including verbs of Caused Accompanied Motion (CAM) has developed as part of the overarching causative schema.

The subschema is formally characterized by the presence of the preposition *a* before the infinitive, a feature that has become completely fixed since the 18th century, and by a low degree of unithood, implying a low degree of syntactic incorporation between the causative verb and the infinitive. In concrete, it has been shown that the postposition of the causee to the infinitive is very infrequent in the subschema with CAM verbs, as opposed to

what was observed for other causatives with *hacer* ('to make') and *poner* ('to put') (Enghels and Comer 2020, p. 176). It follows, therefore, that, compared to related variants, the causative subschema with CAM verbs has a relatively low degree of incorporation.

From a semantic point of view, it has been shown that the subschema with CAM verbs prototypically expressed indirect causation. Moreover, it increasingly specialized in prototypical indirect causation events, involving an inanimate causer and an animate causee. Again, this contrasts with the subschema with *putting* verbs, which specialized in the expression of direct causation, and with the micro-construction with *hacer*, which presents greater semantic versatility (Enghels and Comer 2020).

The two micro-constructions under study also display some diverging characteristics. The micro-construction with *llevar* more rigidly selects inanimate causers than the micro-construction with *traer*, which also admits animate causers with some ease. Moreover, the micro-construction with *llevar* is frequently associated with infinitives expressing perception and communication events, while the micro-construction with *traer* is more strongly associated with attributive events, such as those expressed by *ser* ('to be'). However, most of the grammaticalization parameters analyzed point to a similar degree of grammaticalization of the micro-constructions, which has remained relatively stable over time.

From the perspective of their productivity, the subschema with CAM verbs developed following a S-shaped curve, with a slow beginning followed by a rapid expansion (Nevalainen 2015). However, it has been shown that the micro-construction with *traer* dominated the subschema until the 15th century, while the variant with *llevar* developed later and more slowly. This has been explained by the Aktionsart features of *llevar* as motion verb in Old Spanish, which hindered its incorporation into the causative construction. In contrast, from the 16th century onwards, the consolidation of the deictic meaning of *traer* as motion verb caused the micro-construction with *traer* to become less semantically transparent, and consequently, its productivity decreased. On the contrary, the loss of the original aspectual features of *llevar* made the micro-construction with *llevar* more transparent, so in Classical Spanish, it started leading the subschema. This also explains why after the 18th century the micro-construction with *llevar* expanded to new semantic domains (e.g., caused events of communication), while the micro-construction with *traer* became residual, especially in the 20th century.

The above-mentioned changes illustrate that the semantic changes experienced by *llevar* and *traer* as full verbs determined the productivity of the causative micro-constructions under study. These findings suggest that the full meanings of verbs and their grammaticalized uses are strongly connected in the minds of speakers. Therefore, as Garachana Camarero and Hernández Díaz (2020) already pointed out in their study on verbal periphrases in Spanish, we argue that studies on grammaticalization should pay enough attention to possible diachronic changes in the full lexical meaning of verbs, since they may impact the evolution of grammaticalized constructions.

To conclude, in future research, it would be interesting to study whether the observed tendencies also apply to other causative micro-constructions with CAM verbs, such as *aducir* ('carry') (25a) and *conducir* ('drive, lead') (25b).

(25) a. Por que el entendimiento los **aduxiesse** *a connosçer* las cosas ssegunt que eran primeramiente a Dios (Alfonso X, *Setenario*, 1252–1270).

'That the understanding **lead** them to know things as they were firstly to God'.

b. Esta noticia naturalmente *me* **conduce** *a rectificar* otra. (Benito Jerónimo Feijoo, *Teatro Crítico Universal*, 1734).

'This news naturally **leads** me to rectify another'.

**Author Contributions:** Conceptualization, J.T.S. and R.E.; Methodology, J.T.S. and R.E.; Validation, J.T.S. and R.E.; Formal analysis, J.T.S.; Investigation, J.T.S. and R.E.; Data curation, J.T.S.; Writing—original draft, J.T.S. and R.E.; Writing—review & editing, J.T.S. and R.E.; Visualization, J.T.S.; Supervision, R.E.; Funding acquisition, J.T.S. All authors have read and agreed to the published version of the manuscript.

**Funding:** This research was funded by the Spanish Ministry of Universities grant for university teacher training number FPU19/01505.

**Data Availability Statement:** The research data of this study is available in TROLLing (Torres Soler and Enghels 2023). Torres Soler, Julio, and Renata Enghels. 2023. Replication Data for: From Motion to Causation: The Diachrony of the Spanish Causative Constructions with *traer* ('Bring') and *llevar* ('Take'). DataverseNO. https://doi.org/10.18710/BQ0PDU.

**Conflicts of Interest:** The authors declare no conflict of interest.

## Notes

1    The parenthesis in the slot of the preposition indicates that it may appear or not.

2    In this paper we do not adopt the term *verbal periphrasis*, which many authors use in a restricted sense excluding the causative construction. Instead, we opt for the more comprehensive terms of *construction* and *micro-construction* as understood by CxG. It is assumed, in any case, that we are dealing with verbal expressions that present a certain degree of grammaticalization.

3    This is the most frequent and oldest meaning of the verbs (Torres Soler 2020).

4    The following lines provide a summary of the main ideas exposed in Torres Soler (2023) and relevant to explain the diachrony of the causative micro-constructions under study.

5    An anonymous reviewer correctly warns that this example comes from a document with low ecdotic reliability. Even so, it illustrates in a very clear way the conceptual relationship between the causative construction and the metaphorical expressions from which it derives, and this is why we decided to include the example.

6    See https://www.rae.es/banco-de-datos/cdh (accessed on 15 March 2022).

7    This implies that examples with more than three elements interpolated between the causative verb and the infinitive are not included in our corpus. However, it does not affect our analysis, which aims to clarify whether it is possible or not to interpolate words between the causative verb and the infinitive. The number of interpolated elements is not relevant for the purposes of this work.

8    As other diachronic corpora of Spanish, the CDH contains a much lower number of texts from the 18th century than from earlier and later centuries. Specifically, in the geographic area of Spain it contains 12,372,992 words in the 18th century, whereas the amount of words is 35,471,967 in the 17th century and 39,543,425 in the 19th century. This quantitative data disparity may cause some unexpected results when observing diachronic trends, especially when the number of tokens is low. This will be taken into account during the interpretation of the results.

9    Several arguments have been made in favour of considering *coming* verbs as the marked elements of the pair of deictic motion verbs (Ricca 1993). One of them is the fact that the *going* verb is the one used in deictically neutral expressions or expressions in which the Goal is interpreted in a generic way, as in the expression *me gusta ir al cine* ('I like to go to the cinema'), in which the verb *ir* ('go') can be used even if the speaker is in the cinema at the moment of speech. Another argument is that, sometimes, the use of the *coming* verb generates more rigid inferences about the presence/absence of the speaker at the Goal than the same utterance with the *going* verb. An example is *¿vendrás a la fiesta esta noche?* ('Will you come to the party tonight?'), from which it is inferred that the speaker is at the place of the party at the moment of speech or will be there at the moment of reference, as opposed to the expression *¿Irás a la fiesta esta noche?* ('Will you go to the party tonight?'), from which neither the presence nor the absence of the speaker at the party can be inferred.

10    A similar variation in the prepositional slot has been documented in the causative micro-constructions with poner and meter (Enghels and Comer 2020).

11    Due to the low density of causative constructions in the highly extensive list of search results, we retained only cases in which the causative verb and the preposition *en* or the causative verb and the infinitive were contiguous. Consequently, the frequencies of the prepositional variants with *en* and Ø are not completely comparable to those with the preposition *a*, although they do provide a more concrete idea of their diachronic development.

12    It goes without saying that cases in which the causee is expressed through a clitic are excluded (e.g., *El miedo les llevó a huir* 'fear made them run away').

13    Cases without any adjunct susceptible of being intercalated are excluded from this subsample.

14    The different categorial nature of the causee (NP2) and the adjuncts entails a different degree of syntactic mobility. As a consequence, the comparison of the degree of interpolation of both types of elements with each other does not yield relevant results. Similarly, Comer (2020, p. 355) states that "the grammatical class (nuclear or non-nuclear) of the interpolated constituents

and the degree of grammaticalization do not correlate". Instead, taking the interpolation of different grammatical classes as independent variables allows to test whether the degree of lexical interpolation between the causative verb and the infinitive is subject to diachronic variation or variation across micro-constructions.

[15] Human participants, living animals, institutions (e.g., the Ministry of Universities), and personified divine or mythological beings (e.g., Satan) have been classified as animate. Abstract concepts, objects and nominal expressions designating actions (e.g., a thief's escape) are categorized as inanimate participants. Passive sentences in which there is no explicit causer are excluded (e.g., *Así los niños son llevados a obedecer* 'In this way children are led to obey').

[16] We included the 5 most frequent semantic classes in each century, and only those that appear more than once.

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
