# Peer review of "From Motion to Causation: The Diachrony of the Spanish Causative Constructions with traer (‘Bring’) and llevar (‘Take’)"

_languages, doi:10.3390/languages8020122_

Round 1

Reviewer 1 Report

Excellent paper on a non-explored subject in historical studies on Spanish, namely, the evolution of causative constructions “traer + inf” and “llevar + inf”. The methodology of analysis is adequate to a usage-based approach to language change. The author founds its conclusions in a thorough emptying of one of the best rated historical corpora of the Spanish CdH (Corpus del diccionario histórico), and applies some statistical methods to substantiate its findings. 

I have a few comments and suggestions that I am going to present as a list:

1.     L. 180-186: author points out “traer” changed its meaning as a result of the analogy with “venir”. This statement needs a bit more of information. 

2.     Note 6: Author says that (s)he limits the search of interpolations between the auxiliary verb and the infinitive to three words, because of the limitations of the search options of CdH. In her/his opinion, that is not a problem, as “we do not expect these cases to occur frequently in a corpus”. I suggest the author to point out, instead, that more or less words between the auxiliary verb and the infinitive won’t affect the analysis (s)he did. What is important is if it is possible or not to insert words between auxiliary verb and infinitive. The number is not so relevant. On the other hand, my impression as a Spanish native speaker is that interpolation can admit even 9 words (llevar a la mayoría de los que estamos aquí a admitir que…). Without a deep search in a corpus that not presents the limitations of CdH it is difficult to quantify the frequency of this kind of constructions. However, the fact that they exist, without being unacceptable, could be an indicator that more than 3 words can be inserted in causative constructions.

3.     Section 4. Results obtained in this section are relevant, and they coincide with the periodization of Spanish, as the most relevant changes happened at the end of the Middle Age and from the 18th century ahead. This last chronology must be linked to the called “Primer español moderno” (see Octavio de Toledo 2007, “Un rasgo sintáctico del primer español moderno (ca. 1675-1825): Las relaciones interoracionales con “ínterin” (que)”, en Cuatrocientos años de la lengua del Quijote, Sevilla: Universidad de Sevilla).

4.     Connected to the examples extracted from CdH, I highly recommend changing all these that come from editions ecdotically not trustable, above all those of the Middle Ages. For instance, Las etimologías romanceadas de San Isidoro, text written in the 14th century, but conserved in a copy of the middle of the 15thcentury. The same for “Epístola de San Pablo a los romanos” or Bocados de oro. (see Rodríguez Molina, Javier; Octavio de Toledo y Huerta, Álvaro. «Acceso a CORDEMÁFORO». Scriptum digital. Revista de corpus diacrònics i edició digital en Llengües iberoromàniques, 2017, 6, p. 69-69). Also, we recommend not to include books by Juan Fernández de Heredia, as he was from Aragon. 

5.     Finally, due to the tie affinity of causative constructions under study to the interpolation of the cause between “traer/llevar” and the infinitive, I wonder if these constructions are not specialized in this specific pattern. In fact, one of these verbs, “llevar”, is grammaticalized in a verbal periphrasis that prototypically follows the pattern “llevar + temporal complement + gerund”. 

Author Response

The response to reviewer 1's comments is attached.

Reviewer 2 Report

Since the overall assessment of the article is positive and its publication is recommended with minor changes, I will point out below only those aspects that, in my opinion, need to be improved. In many cases these are only requests for clarifications. The comments are listed in order of pagination (line number is indicated):

(a) General comment: I wonder if it might not be convenient to present the data not by centuries, but grouped by periods: Old Spanish, Classical Spanish, Modern Spanish, Contemporary Spanish. Perhaps making these groupings would allow some additional generalization. If this is not done, it should be explained why this decision was made.

b) In the Introduction (lines 36 to 71) causative constructions with "meter", "poner" and "hacer" are mentioned. Please provide an example of these constructions to help the interested reader who is not a native speaker of Spanish to understand them.

c) In lines 75-76 and 643 it is pointed out that the microconstructions studied, especially the microconstruction with "llevar", express indirect cause. This is taken for granted, following the works of Da Silva 2004 and 2012. The paper argues that the empirical findings (e.g., animacy of causer and caused) are compatible with this characterization but no independent evidence is offered to show that this is the meaning of the construction. I think this part needs to be clarified and developed a bit with independent arguments as to why these structures express indirect causation.

d) Sections 1 and 2 bias the reader into thinking that the existence of a process of grammaticalization of the microconstructions under study is going to be defended, but this is not the case. Finally, the existence of a process of semantic specialization that is not accompanied by a process of syntactic grammaticalization is defended. If I have understood this idea correctly, then, it would be better that in section 1 or 2 the authors clearly develop the idea that the diachronic pattern of a construction can lead to either grammaticalization or specialization, and that they already anticipate what is going to be defended with respect to microconstructions with "llevar" and "traer".

e) From line 179 onwards, explanations are given on the aktionsart of the events that can accompany CAM verbs and on their deictic properties, which are not illustrated with examples in all cases. It is necessary to add examples for all the statements made (even if only one) to help the reader follow the argumentation. This seems to me absolutely necessary to make this section adequately clear.

f) In the same vein, it would be very convenient to add at the end of section 2 (line 254) a summary table comparing the eventive properties of the two verbs "llevar" and "traer" in relation to their meaning evolution. These different properties and evolution will then be used to construct arguments later in the article (line 367 ff), so it is convenient that the reader can return to this summary table.

g) Lines 306-317. Please see comment (c), as the same applies to this paragraph.

h) The italics in the verbs in footnote 9 are missing.

i) Sections 4.2.2 and 4.2.3 An explanation should be added as to why there is such a difference in the behavior of "causees" and adjuncts, if interpolation of both elemens is used as an argument for the existence of Incorporation.

j) As mentioned in the article, one of the tests used in causative constructions to detect indirect causation is the tempo-spatial separation of events by means of different temporal adjuncts. Have no examples been found  in the corpus that could illustrate this point?

k) In Table 8, in the penultimate row,  I think the "+" is a typo in [+anim].

l) Lines 656-816. In this part of the article different empirical generalizations about the constructions are presented, but sometimes they are oriented from the expectation that there was a grammaticalization process and sometimes from the claim that there is not such a process. This is confusing for the reader. Please take this comment and comment (d) above into account and give the reader a clearer perspective of which diachronic process is to be defended as existing, so that the generalizations are stated consistently from that point of view (there is specialization of meaning but no grammaticalization in the strict sense, if I have not misunderstood).

m) In lines 811-812 it is said that "both micro constructions were at some point linked to attributive caused events, as it was expected because of their origin". If these verbs were originally verbs of motion, this statement is not understood. Please clarify.

Finally, the examples need to be formatted. I note this here although I understand that this will be done at a later stage.

Author Response

The response to reviewer 2's comments is attached.
